# Modelling the Mobility Changes Caused by Perceived Risk and Policy Efficiency

**Sijin Wu** [1,2] ⬤, **Susan Grant-Muller** [2] **and Lili Yang** [1,*]

1   Department of Statistics and Data Science, Southern University of Science and Technology,
    Shenzhen 518055, China
2   Institute for Transport Studies, University of Leeds, Leeds LS2 9JT, UK
*   Correspondence: yangll@sustech.edu.cn

**Abstract:** In many countries, governments have implemented non-pharmaceutical techniques to limit COVID-19 transmission. Restricting human mobility is one of the most common interventions, including lockdown, travel restrictions, working from home, etc. However, due to the strong transmission ability of the virus variants, further rounds of interventions, including a strict lockdown, are not considered as effective as expected. The paper aims to understand how the lockdown policy and pandemics changed human mobility in the real scenario. Here we focus on understanding the mobility changes caused by compliance with restrictions and risk perceptions, using a mobility index from the Google report during three strict lockdown periods in Leeds, the largest city in the county of West Yorkshire, England, from March 2020 to March 2021. The research uses time-varying z-scores and Principal Component Analysis (PCA) to simulate how local people dynamically process and perceive health risks based on multi-dimensional daily COVID-19 reports first. Further modelling highlights exponentially increasing policy non-compliance through the duration of lockdown, probably attributable to factors such as mental anxiety and economic pressures. Finally, the proposed nonlinear regression model examines the mobility changes caused by the population's dynamic risk perceptions and lockdown duration. The case study model in Leeds shows a good fit to the empirical mobility data and indicates that the third lockdown policy took effect much slower than the first. At the same time, the negative impact of the epidemic on population mobility decayed by 40% in the third lockdown period in contrast with the first lockdown. The risk perception estimation methods could reflect that the local population became increasingly accustomed to the COVID-19 situation, and local people rationally evaluated the risks of COVID in the third lockdown period. The results demonstrate that simulated risk perceptions and policy decay could explain urban mobility behaviour during lockdown periods, which could be a reference for future decision-making processes.

**Keywords:** urban mobility; dynamic risk perception; data-driven model; policy analysis

## 1. Introduction

With the emergence and spread of the COVID-19 virus pandemic worldwide, governments have imposed intensive non-pharmaceutical interventions (NPIs) on human mobility and social activities whilst effective vaccines were developed and distributed, which are considered as crucial methods at the beginning of the pandemic to control virus transmission rates [1]. However, the continued challenge is that some virus variants might break the vaccination protection and have super transmission rates, which may be difficult to control immediately [2]. In the U.K., people experienced three separate strict lockdown periods in a year from March 2020 to March 2021 to limit the peak of virus transmission and protect public health capacity. Facing over 100,000 new confirmed cases per day in December 2021 in the U.K. [3], the government again considered and implemented restriction policies. Nevertheless, existing evidence on mobility recovery during the lockdown showed a decreasing effect in reducing human mobility even when daily cases were rising,

which means the lockdown policy could not be considered a sustainable, efficient NPI in the long term [4–6]. This demands that policymakers should evaluate the relationships between implemented NPIs and human mobility, because changes in mobility patterns to mitigate health impacts and in response to national regulation may become long-term rather than transient, thereby impacting planning decisions for the built environment and for inter-urban connectivity. Thus, understanding how and why the real-world mobility behaviour changed during the lockdown implementation could help stakeholders make improved decisions and reduce unnecessary costs for sustainable development [7–10].

The existing literature indicates that the perception of risk by the public towards the environment is a critical factor influencing human mobility. Many pieces of research have stated a positive correlation between the implementation of lockdown policy and a reduction in mobility in different countries or regions [11–13] through reported cases and statistics, such as those in the form of a mobility index. Some research fully explores the relationship between mobility and risk perception related to COVID-19. Across survey-based research in ten countries, risk perception is regarded as the critical feature correlated with the reported public health compliance [14]. Nelson et al. gathered online surveys and detected a positive correlation between the concern of COVID and self-quarantining behaviour in the U.S., Canada, and Europe at the start of the outbreak [15]. Chan et al. found that areas with high risk-tolerance are positively associated with the mobility change in retail and recreation places [16]. In addition, Harrison et al. discussed the influence of different transport modes on a perception of safety by building the Causal Loop Diagram of a broader transport-health system [17].

However, to the authors' knowledge, currently published research on the quantitative relationship between mobility and risk perception in the population and lockdown policy is limited in two respects. First, the longitudinal risk perceptions towards COVID-19 are hard to collect or record in traditional survey methods. It is known that individual perceptions and behaviours may change with the evolving environment regarding COVID-19 severity, policy, and infrastructure implementations [18]. But the main research methods used to monitor public risk perception are questionnaires, which make consistently tracking the longitudinal nature of risk perceptions and mobility behaviours hard. Much existing analysis relies on a point in time or some periods during the pandemic period, which will lose track of the evolving mobility and changes in risk perception. For example, Wise et al. tracked participants for over a week (11 March 2020–16 March 2020), beginning at the start of the pandemic in the U.S., to discuss the association between risk perception and commitment to protective behaviours [19], but this study duration is too short of building a dynamic analysis. Another study surveyed travel risk perception and travel behaviour in the Germany-Austria-Switzerland region in March 2020 and two weeks later as two separate periods [20]. The research only compares the changes in two periods. The most recent referenced longitudinal study of risk perception in the U.K. tracked different participants using the same cross-sectional variables for five time-snapshots over ten months. Even though the results indicate that risk perception is a dynamic process and positively correlated with health behaviours [21], the analysis is based on limited time points rather than the daily updated dynamics. On the other hand, this work satisfied the longitudinal definition but failed to model the dynamic relationships between mobility and risk perceptions for the specific population, which induces the second research gap in our study. Most mobility research relies on correlative analysis or empirical statistics that only represent the association relationships and cannot model the quantitative influence or directionality between mobility and risk perceptions, while the existing research implies the risk perception toward COVID-19 acts as an antecedent of behavior [22]. For example, Nouvellet et al. linked mobility at the national level to effective reproductive numbers to examine the relationships between mobility and severity [23]. The results suggested that lockdown policies have a diminishing impact on the control of the pandemic in many countries but could not provide a quantitative analysis. Similarly, Joshi found that the lockdown policy has decreasing effects on human mobility through empirical analysis

without quantifying the possible influencing factors [24]. Kraemer et al. visualised the extracted mobility data in China in a timeline with a lockdown indicator, showing that the statistical correlation between the number of cases and mobility dropped from positive to negative after implementing the control measures without considering the self-risk perceptions [11]. In conclusion, the quantitative model or mechanism explaining why the efficiency of lockdown restrictions is fading and how risk perception affects mobility is not explored as yet due to many reasons. Hence, it is meaningful to determine and explore how health risk perceptions and policy affect and interact with health-protecting behaviours.

To address the above research gaps concerning the lack of longitudinal data supporting the quantitative modelling, the current research proposed the algorithms to estimate the dynamic COVID-19 risks perceptions incorporating the daily COVID-19 reports, including local cases, local deaths, national cases, and national deaths as the input. The research also used the Google Community Mobility Report for retail and recreational facilities in Leeds from March 2020 to June 2021 as the mobility index, which covered three strict lockdown periods, as the dependent variable data. These two data sources can form panel data that record the population's mobility behaviour and risk perception of the pandemic because they match the need for daily consistent collection frequency and focus on the population in the same local area. This is not a strict experimental panel however, as the location population will be subject to some movement, temporary visitors etc. Another advantage is that retail and recreational mobility can reflect the population's tendency for policy non-compliance because it is defined as mobility trends for non-essential retail places such as restaurants, cafes, shopping centres, theme parks, museums, libraries, and movie theatres. These places were normally not open to the public during three strict lockdown periods. Hence, this mobility index only records the non-essential mobility and excludes mobility behaviours concerning essential activities such as visiting food stores.

To dynamically model mobility behaviour during lockdowns, the proposed regression model considers the two most significant perspectives influencing mobility behaviours and policy compliance: confirmed cases and personal psychological characteristics, based on evidence and suggestions from previous research in the U.K. [21]. The originality of the research is the use of a time-varying z-score and PCA to estimate how people perceived the pandemic severity through multi-dimensional daily reports and history information; results can be validated by satisfying the general trends of risk perception in the U.K. [21]. The proposed model also quantitatively describes that the non-compliance for lockdown increased exponentially as the policy continued over time, which is consistent with the survey results during the first lockdown in the U.K. [25]. Finally, the method sheds light on how daily reports and lockdown duration quantitively affected population mobility during the lockdown and how they could be utilised to predict mobility recovery trends.

## 2. Materials and Methods

This section firstly introduces the data used in this research and how we process the data before fitting the model. Then, the methods for generating the risk perception and policy non-compliance are presented. Finally, the nonlinear regression model is presented, which describes how modelling factors influence mobility.

### 2.1. Data

The mobility index used in this study is derived from Google's COVID-19 Community Mobility Report; data are collected from Google users who have turned on their location history (Google LLC *"Google COVID-19 Community Mobility Reports"*. https://www.google.com/covid19/mobility/ accessed on 10 December 2021), and this work further focuses on the city of Leeds, U.K, as a case study. In the literature, the dataset has been used to analyse land-use and economic activity in different regions during the COVID-19 pandemic [26,27]. It is also helpful for understanding the spatiotemporal patterns of the effectiveness of policy intervention on mobility [28], and other restrictions and guidance such as social distancing [29]. Google's COVID-19 Community Mobility Report

and other similar products (mobility index by Apple) can provide bedrocks for evaluating epidemic trends and policies; Corentin et al. utilised different mobility indexes to evaluate the correlations between social distance and epidemic trends in states of the U.S. [30].

The mobility index corresponding to "the mobility changes for each day of the week" is continuous in terms of percentage relative to the median of each specific day during the baseline period from 3 January to 6 February 2020 (approximately seven weeks before the introduction of the first national lockdown). There are six mobility categories: retail and recreation, groceries and pharmacies, parks, transit stations, workplaces, and residential. We chose to focus on the mobility index of retail and recreation places as it is directly relevant to the lockdown as the key element of the policy was the closing of all non-essential shops and services. This category includes places such as cinemas, restaurants, shopping centres, etc. It is worth mentioning that supermarkets, groceries, and pharmacies were not included in this category by the Google Mobility Report [31], and are regarded as necessary shopping for lives. Therefore, the retail and recreation mobility can represent the effectiveness of the lockdown policy more precisely.

Another data used in this study are the official COVID-19 statistics, which were obtained from the U.K. Coronavirus Dashboard website (https://coronavirus.data.gov.uk/ accessed on 10 March 2022). The website provides daily updates about the local cases, local death numbers, national cases, and death toll. The vaccination statistics are available from January 2021, but are not included in this study and models because most lockdown periods were before the large-scale vaccine distribution.

In summary, the research data consists of two parts: the mobility index and the COVID-19 reports. The research approach is outlined in Section 2.2 and generates dynamic risk perceived by daily COVID-19 reports as one of the independent variables. The dependent variable is the mobility index of retail and recreation places. The specific data pre-processing and introduction is explained in Section 3.1.

### 2.2. Dynamic Risk Perception towards COVID-19 Estimated by Time-Varying z-Score

Dynamic risk perceptions are required to model the continuous changes in mobility, which are challenging to collect directly with traditional surveys. Hence, the research referred to the time-varying z-score to simulate how people process the COVID-19 information and perceive the risk. Since the mobility data was collected from peoples' mobile devices, the research assumes that the users of these devices can easily access daily COVID-19 reports, including local daily cases, local daily deaths, national daily cases, and national daily deaths from the media with a reliable probability. In practice, information on these reports was proactively and widely publicised, with high visibility, through the various media. There were multiple information channels with prominent COVID-19 daily reports, such as T.V. news channels, online and hard copy newspapers and more. These figures and information will have affected the public's perception of risk towards COVID-19. For instance, mobile phone users received subscribed news summarising COVID-19 reports from the previous day. The users will have perceived and processed the latest information and compared it with the historical information, then today's risk perceptions against COVID-19 were assessed. The following explanatory factors (variables) were considered to influence the dynamic risk perception:

- Local daily cases
- Local daily death numbers
- National daily cases
- National daily death numbers

These four factors are all daily updated indexes, and the cumulative death or cases are excluded because they have high collinearity with the time (see Appendix A.1). The order of magnitudes is different for each selected variable due to the volume sizes between local and national data, especially for national daily cases and local daily deaths. Furthermore, risk perception in the psychological scale is often conducted on the 5-point Likert scale [14,21]. It is also important to normalise the variables to compare measurements with different units. Therefore, variables measured on different scales do not contribute equally to the

analysis, and it is necessary to re-scale the variables. One helpful data processing method is the standard scaler (i.e., z-score), making the variables with zero-mean and unit variance. The z-score has been used widely in a variety of applications and contexts, a specific example being in economic research as a risk measure that reflects a bank's likelihood of bankruptcy [32–34]. The advantage of using this approach is that, if the data is sequenced in order of time, it is possible to analyse the data according to the most recent set of observations, or to give different weights, exploring z-score variants to track dynamic changes in the data, which is the so-called time-varying method. The time-varying z-score can be calculated by adjusting the moving mean and standard deviation with whole history samples or given window size [29–31].

This research tested three different methods to calculate the time-varying z-score for each variable's time series. According to the definition of z-score calculation, the adjustments were applied in the calculation methods for the sample mean and standard deviation. For example, the local daily case is a sequence $\{X_t\}$, $t$ from 1 to $T$, where $T$ is the lockdown duration. The three proposed algorithms use the moving average and standard deviation with a specific window size (Algorithm 1: moving z-score with window size n (MZ_n)), the moving average and standard deviation with all history information (Algorithm 2: moving z-score with all history information (MZ_all)) and the exponentially-weighted moving average/standard deviation (Algorithm 3: exponentially weighted moving z-score with window size n (EMZ_n)) to calculate the moving z-score respectively. After accessing the datum $\{X_t\}$ in the different algorithms, there different dynamic risk perceptions were generated by the local daily cases.

---

**Algorithm 1** Moving z-score with window size n (MZ_n)

---

$Input\ \{X_t\}, N$
$X_0 = 0$
$For\ t\ in\ 1, 2, \dots, T:$
　　$IF\ t < N:$
　　　　$\overline{X_t} = X_t$
　　　　$se_t = 0$
　　$ELSE:$
　　　　$\overline{X_t} = \frac{\sum_{t-N}^{t-1} X_i}{N}$
　　　　$se_t = \sqrt{\frac{1}{t-1} \sum_{t-N}^{t-1} \left(X_i - \overline{X_t}\right)^2}$
　　$IF\ se_t = 0:$
　　　　$R_t = 0$
　　$ELSE:$
　　　　$R_t = \frac{X_t - \overline{X_t}}{se_t}$
$Output\ \{R_t\},\ t = 1, 2, \dots, T$

---

**Algorithm 1.** Dynamic perceived risk perception by moving average and moving standard deviation with window size as N.

---

**Algorithm 2** Moving z-score with all history information (MZ_all)

---

$Input\ \{X_t\}$
$X_0 = 0$
$For\ t\ in\ 1, 2, \dots, T:$
　　$\overline{X_t} = \frac{\sum_1^{t-1} X_i}{t-1}$
　　$se_t = \sqrt{\frac{1}{t-1} \sum_1^{t-1} \left(X_i - \overline{X_t}\right)^2}$
　　$IF\ se_t = 0:$
　　　　$R_t = 0$
　　$ELSE:$
　　　　$R_t = \frac{X_t - \overline{X_t}}{se_t}$
$Output\ \{R_t\},\ t = 1, 2, \dots, T$

---

**Algorithm 2.** Dynamic perceived risk perception by moving average and moving standard deviation with all historical samples.

---

**Algorithm 3** Exponentially-weighted moving z-score with window size n(EMZ_n)

---

$Input\ \{X_t\},N$
$X_0 = 0$
$\alpha = 2/(N+1)$
$\mu_0 = X_1$
$For\ t\ in\ 1, 2, \ldots, T:$
　　$\delta_t = X_t - \mu_{t-1}$
　　$\mu_t = \mu_{t-1} + \alpha * \delta_t$
　　$\sigma_t = (1-\alpha)\left(\sigma_{t-1} + \alpha * \delta_t{}^2\right)$
　　$IF\ \sigma_t = 0:$
　　　　$R_t = 0$
　　$ELSE:$
　　　　$R_t = \frac{X_t - \mu_t}{\sigma_t^{0.5}}$
$Output\ \{R_t\},\ t = 1, 2, \ldots, T$

---

**Algorithm 3.** Dynamic perceived risk perception by exponentially weighted moving average and exponentially weighted moving standard deviation with window size as N.

For Algorithm 1, the research uses $N = 7$, 14, and 28 to represent the population's short/medium/long-term memory when producing the risk perception from daily reports. For Algorithm 2, since all historical data are deemed equal, the calculated risk perception represents the long-term memory of the population's risk perception. As for Algorithm 3, even though every historical sample is considered, the weight of each sample decreases exponentially from the earliest to the oldest. The $N = 71,428$ in Algorithm 3 represents that the first N datum points represent about 86% of the total weight in the calculation when $\alpha = 2/(N+1)$.

Recall that the variables considered form four sequences: local case, local death, national case, and national death. To avoid the dimension explosion fitting the model (see Appendix A.1) and simulate the population considering the four variables together, the last step for generating the perceived COVID-19 risk was Principal Component Analysis (PCA). The complete process of generating the perceived risk perception by daily reports is shown in Figure 1. The four variables used the same algorithm to generate the risk perceptions and then be compressed (through PCA) into one dimension as the final variable that was used to represent the intraday risk perceptions toward COVID-19 influencing mobility.

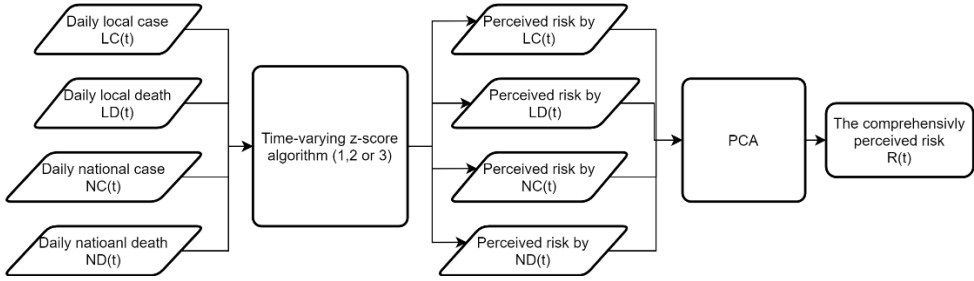

**Figure 1.** The perceived risk generation process.

## 3. The Case Study and Results

The case study area in this work is Leeds, U.K. Leeds is the largest city in West Yorkshire, which has a population of 793,139 (mid-2019 est.) [35]. It has one of the busiest railway stations and important motorway links in the North of England and is famous for its many shopping arcades and diverse economy. Using the Ordnance Survey's classification of Point of Interests (POI), the research recorded each POI according to the definition of categories in Google Mobility Report [31]. They are illustrated in the map in Figure 2.

The cyan region is the study area (Leeds), and each orange point represents a retail and recreational place, whose mobility changes are the focus of this research and reflect the policy non-compliance and risk perceptions during the lockdown. The spatial distribution of varying types of POI (Figure 2) also reflects the urban form, functions, and land-use. The retail and recreational places are clustered in the city centre, while workplaces are mainly located in the urban periphery, with public transitions distributed along the roads. However, the spatial patterns are static and lack a temporal dimension to the mobility changes. Hence, this research implements the dynamic risk perceptions and policy duration to study the changes in mobility in Leeds during the COVID-19 epidemic.

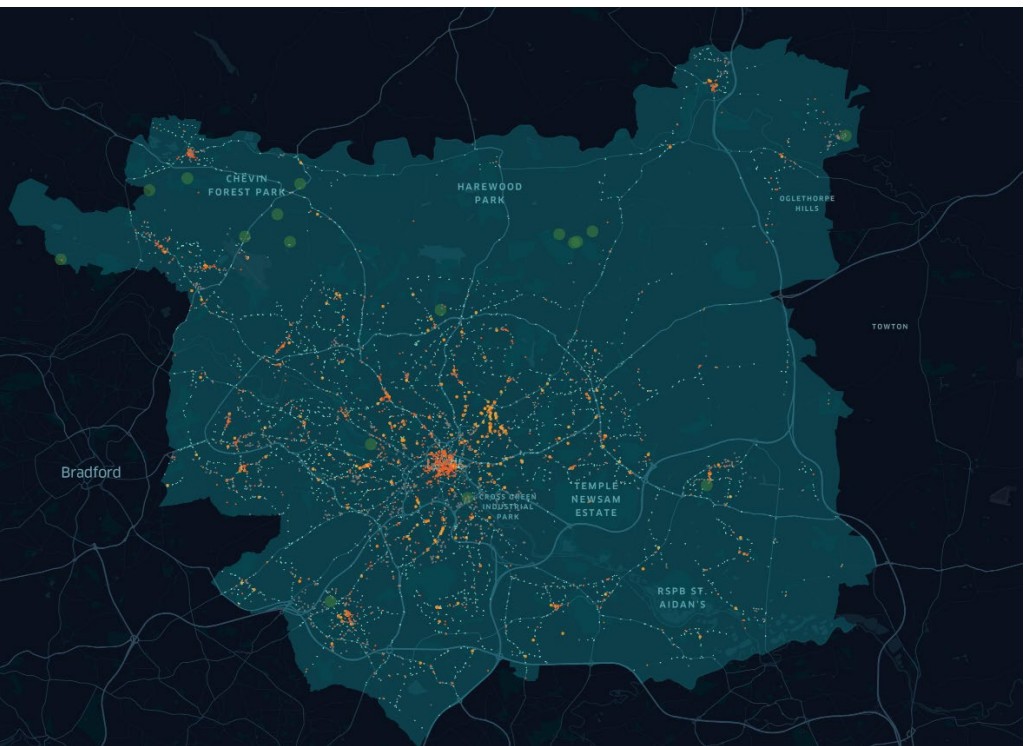

**Figure 2.** The Points of Interest in Leeds.

An advantage of focusing on a single city is that this could avoid exogenetic factors on mobility, such as divergent policy interventions in different local authorities. For example, different cities have various policy restrictions, population distributions, weather, and health care capacity. Hence, we only considered the mobility inside the case study area rather than a larger region. The approach as a whole is transferable to other cities for which similar data is available, however.

Section 3.1 introduces the data pre-processing. Next, Section 3.2 presents the formulas describing how mobility is affected. Finally, after testing the statistical results of combinations of algorithms and formulas, the optimally fitted models are stated in Section 3.3.

### 3.1. Data Pre-Processing for Mobility Index

Google's COVID-19 Mobility Report presents the movement trends since 15 February 2020. Mobility is measured based on the frequency and duration of visits to places, as well as calculating popular times in Google Maps. Using a percentage format, the Google report data describes the mobility change compared with the median value from the baseline period from 3 January to 6 February 2020. If values are negative, the mobility on this date is less than the baseline scenario, and vice versa. The trends are recorded and classified into different categories: retail and recreation, groceries and pharmacies, parks, transit stations, workplaces, and residential. The data is open for downloading and research. One apparent observation is that curves have periodic patterns [36]. To visualise the periodicity directly,

the author sliced the section between 23 March and 23 June 2020, this being the period from the first strict lockdown in the U.K. to the end of national hibernation announced by the government, in Figure 3.

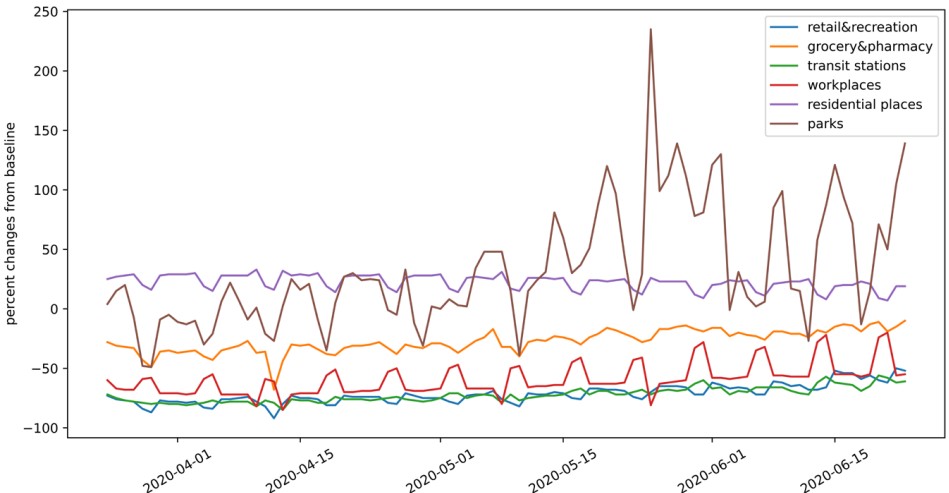

**Figure 3.** The mobility trends during the first lockdown in Leeds.

All curves are almost flat in the first few weeks, and then the mobility for the park shows significant vibrations and an increasing trend. In addition, the workplaces and residential places exhibited complementary periodic patterns because of the weekdays and weekends. Since the policy strictly limited the indoor social activities and business service industries, the retail and recreation places should have had significant mobility loss or keep at a low-level. However, by decomposing the time series, the increasing trend is apparent by showing the mobility trend for retail and recreation places in Figure 4. This is counterintuitive because the lockdown restrictions were not lifted during these months, and the accumulative deaths and cases of COVID-19 were increasing. The possible reasons could be people's risk perception toward COVID-19, and natural policy decay over time.

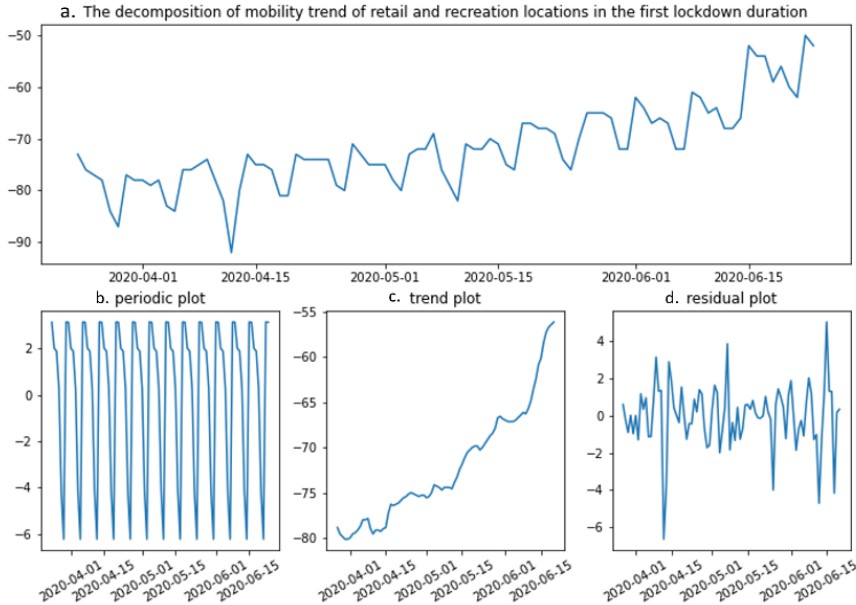

**Figure 4.** The decomposition of retail and recreation mobility during the first lockdown period in Leeds. According to the additive model in time series, the original mobility curve (**a**) is the sum of three components: seasonal component (**b**), trend component (**b**,**c**), and random (**d**) curves.

Figure 4 shows an evident weekly pattern in the original mobility index. The data is further aggregated based on the day of the week, and a boxplot is shown in Figure 5. Hence, this research smooths the mobility data by using the moving average method with a window size of seven days to smooth out the one-week periodicity.

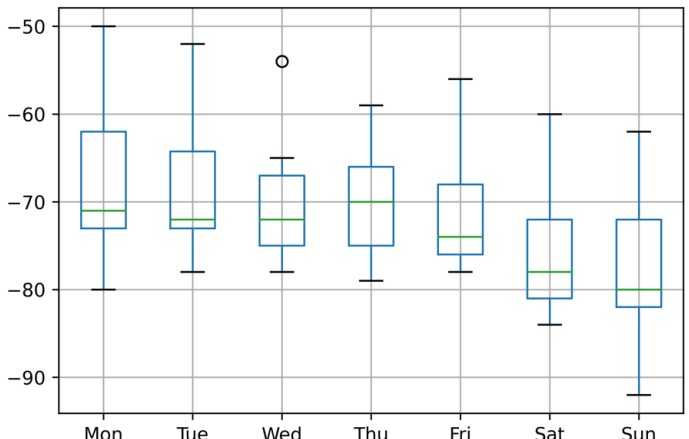

**Figure 5.** The mobility boxplot of each weekday. The median and Interquartile range (IQR) on weekdays are higher than on weekends, implying that weekly patterns in mobility are closely associated with local lifestyles.

The effect of the moving average is to smooth the curve and decrease the impact of the noise fluctuations on the model. Furthermore, the experiment results in the Appendix A (see Appendix A.2) show that the model fitted with rolling mobility data has better evaluations than raw mobility data. In conclusion, for the dependent variable (mobility index), the moving average with window 7 of mobility will smooth the objective curve and improve the model fitting performance.

### 3.2. Model Configuration

This section implements a detailed case study to analyse the retail and recreational mobility changes among three national strict lockdown periods in Leeds from March 2020 to March 2021. Time-series of mobility changes are shown in Figure 6, and the complete COVID-19 rules are summarised in Figure 6 from the authority source [37].

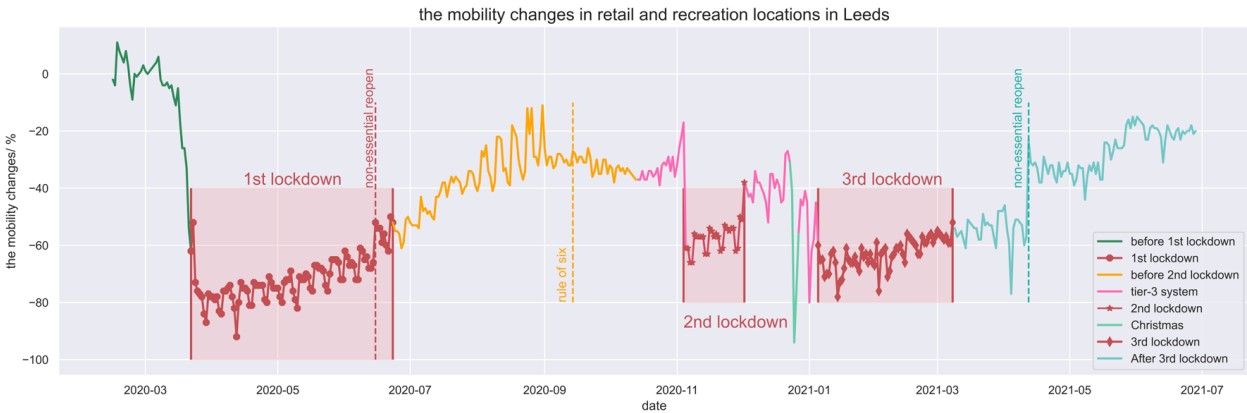

**Figure 6.** The mobility changes of retail and recreation locations in Leeds. The first lockdown started on 23 March 2020 and ended on 23 June 2020. The second lockdown lasted one month, from 31 October 2020 to 2 December 2020. The last lockdown was from 6 January 2021 to 8 March 2021.

The research implemented three algorithms with different window sizes to generate the dynamic perceived risk $R(t)$ and to model the local aggregated mobility changes in each

lockdown period. The $R(t)$ developed by various methods is validated and discussed in the Appendix A (Appendix A.2). Since the objective was to find trends in mobility behaviours influenced by policy and risk perceptions, the mobility index data was smoothed by applying the centred 7-day moving-average method to eliminate the weekly periodicity mentioned in Section 3.1. The research focuses on two independent variables: the policy duration, $t$, and dynamic risk perceptions, $R(t)$. The regression model is based on the general linear model as

$$M_t = c + \beta_1 \varphi_1(t) + \beta_2 \varphi_2(t) + \ldots + \beta_n \varphi_2(t) + \alpha R(t), \ t \in (1, T) \tag{1}$$

where $T$ is the lockdown duration and $R(t)$ is the dynamic risk perception generated by the proposed algorithms in Section 2.2.

Under full compliance, the assumed lockdown policy effect was that mobility should drop to a low level and stay there, which is inversely proportional to the lockdown time. The first regression term is assigned as $\frac{1}{t}$. However, the real situation in some lockdown periods of Leeds reviewed was that after a sharp drop in the beginning, mobility recovered at a slower rate throughour the lockdown days. The proposed model assumes that the mobility recovered at a fixed rate, which assigns the second term $\varphi_2(t) = t$. In conclusion, the first proposed linear regression model is

$$M_t = c + \beta_1 \frac{1}{t} + \beta_2 t + \alpha R(t), \ t \in (1, T) \tag{2}$$

Furthermore, according to Ganslmeier's work [25], the lockdown policy's non-compliance increased at an exponential rate in the U.K. Hence, the alternative model is

$$M_t = c + \beta_1 \frac{1}{t} + e^{\beta_2 t} + \alpha R(t), \ t \in (1, T) \tag{3}$$

which is a nonlinear model.

The final case study regression models for each lockdown period were selected by comparing the goodness-of-model fitting and on the basis of sensible signs for the parameters. In total, we have seven kinds of perceived risk generated by three proposed algorithms and window sizes. The window size is chosen as 7, 14, 28 and all historical data representing various memory capacities. The objective is to compare the model results and search for the proper risk perception feature in each lockdown period and which modelling term, linear or nonlinear, is more precise. All experiment records are available in Appendix A (see Appendix A.3).

### 3.3. Model Results

The current research compares the series of experimental results to systematically select the model type and parameters that fit the data optimally. In total, there are two optional model types, linear and nonlinear, discussed in Section 3.2. Three lockdown periods are fitted in different algorithms and parameters for each model type. A correct, well-fitting model should have statistical significance for each parameter, especially the estimated parameter $\beta_2$ which should be negative, keeping consistency with other researchers' conclusions [21,22], which means risk perception should negatively affect mobility. After implementing a range of experiments and comparisons (see Appendix A.3), we could finally select well-fitting models that properly describe the mobility changes in three periods. The estimated regression model with the best performance for the first lockdown period is nonlinear:

$$M_t = -82 + 29.56 \frac{1}{t} + e^{0.03t} - 0.51 R_{all}(t), \ t \in (1, T_1) \tag{4}$$

where $R_{all}(t)$ is estimated by using all historical information (Algorithm 2) and $T_1 = 91$ is the length of the first lockdown policy. It is reasonable because people were nervous about

the unknown risk and virus due to the beginning of the pandemic. Hence, people tended to hold long-lasting memories of the COVID-19 spread.

And the selected regression model for the third lockdown is also nonlinear because of its better goodness of fit:

$$M_t = -68.69 + 9.55\frac{1}{t} + e^{0.04t} - 0.3R_{ew}(t), \ t \in (1, T_3) \tag{5}$$

where $R_{ew}(t)$ is estimated by using the exponentially weighted moving average/standard deviation with $N = 28$ (Algorithm 3) and $T_3 = 62$ is the length of the third lockdown policy. This implies that people would use the recent information to evaluate the COVID-19 severity and risk, rather than all historical information.

However, for the second lockdown period, none of the proposed models could reach statistical significance for parameter $\alpha$. The regression results based on linear and nonlinear models had almost the same effect. To be comparable with the other two models, the selected model was also chosen as nonlinear for sketching the mobility changes in the second lockdown in Leeds:

$$M_t = -62.23 + 35.64\frac{1}{t} + e^{0.06t} - 0.29R_{all}(t), \ t \in (1, T_2) \tag{6}$$

where $R_{all}(t)$ is estimated by using all history information (Algorithm 3) and $T_2 = 28$ is the length of the second lockdown policy.

The three model fitting results with the best fitness in three periods are illustrated separately in Figure 7. From the perspectives of MSE (Mean Squared Error) and R square, the models can capture the fundamental trends in each lockdown period.

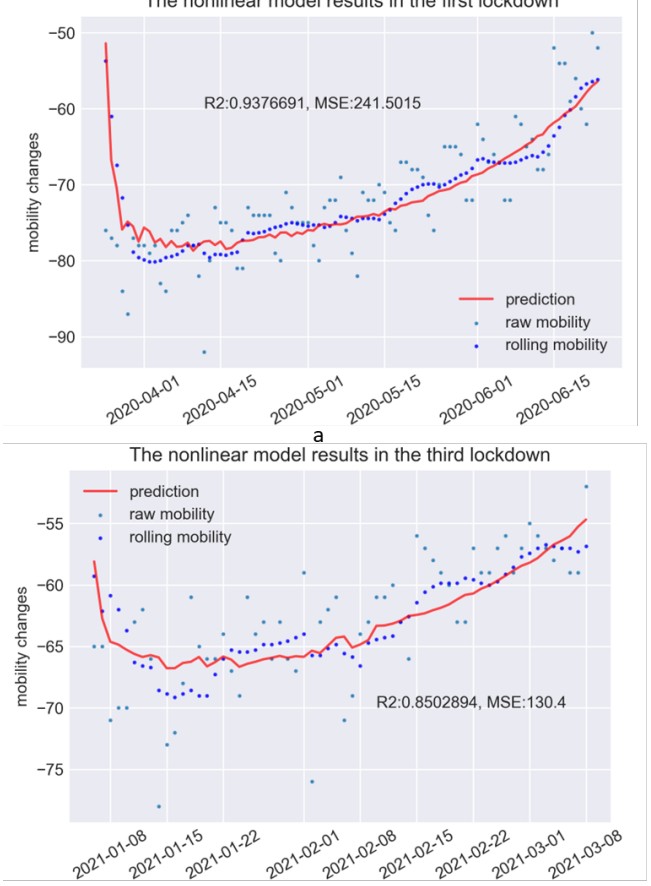

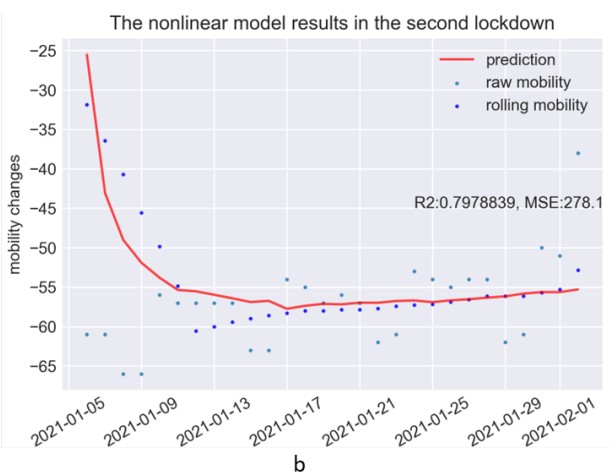

a.  The nonlinear model results in the first lockdown
b.  The nonlinear model results in the second lockdown
c.  The nonlinear model results in the third lockdown

The red line is the predictive mobility. The blue points represent the mobility after applying the 7-day moving-average. And the cyan points are the raw mobility scatter plot.

**Figure 7.** The nonlinear model results in each lockdown period.

By comparing the parameters among three lockdown periods from equation 4 to 6, the changes in parameters can be interpreted as the evolving process of public attitudes toward COVID-19 and lockdown policy. The parameter $\beta_1$ represents the rate of policy efficiency, i.e., the decay rate of the mobility index. The values of $\beta_1$ increased from 29.56 in the first lockdown to 35.64 in the second lockdown and then dropped back to 9.55, implying that the policy took effect in different lockdown periods. In the long term, the lockdown efficiency in the third lockdown was far lower than at the beginning stage, explaining the lockdown fatigue phenomenon. Parameter $\beta_2$, which can be interpreted as the policy non-compliances sensitivity of lockdown time, has the same trend as $\beta_1$. In the first lockdown period, $\beta_2$ was 0.03 but rose to 0.06 in the second lockdown. The potential reason could be the easing of lockdown [5] and the high demand related to shopping and friends gathering to celebrate the Christmas holidays. However, in the third lockdown, $\beta_2$ dropped back to 0.04. This could be due to more and more people realising the importance of lockdown or getting used to lockdown life during COVID-19. The parameter of risk perception's effect, $\alpha$, parameter of risk perception's effect, decayed from 0.51 to 0.3, almost a 40% drop, indicating that people were less affected by COVID-19 cases than in the initial epidemic. The possible reasons could be that, firstly, people were becoming familiar with COVID-19, with the virus appearing to have different levels of impact in different parts of the population. Whilst some population sub-groups continued to have high fatalities, others had much lower fatality rates, particularly in the case of the younger population [38]. Secondly, the vaccine project started in Dec 2020. Many older people (a high-risk group) were vaccinated and protected at that time. It is worth noting that the dynamic risk perceptions in the first two lockdown periods, equation 4 and 6, used Algorithm 2 (i.e., using all historical information to perceive current risks), and Algorithm 3 was more suitable in the third lockdown (Equation (5)). It suggests that after coexisting with the virus for a year, the local population's travel choices were not as influenced by their long-lasting memory of former situations, and that they were focused on the current scenes when considering the risks.

The best linear model used the risk perception generated by the 28-day moving average and standard error for the first lockdown. However, the parameter of the risk perception was non-significant ($p = 0.22$), which was not as good as the nonlinear model ($p = 0$). Using the risk perception generated by all history information, the nonlinear model had the best performance. It can be interpreted that most people had a strong protective mind at the beginning of the epidemic and were cautious due to the uncertainties. Therefore, the public will consider all the history information to judge the current risk, which is not too long to remember. Additionally, as for the policy non-compliance term ($e^{0.03t}$ vs. $0.26 * t$), the exponential term had a consistent conclusion with the previous survey results [39], which means the non-compliance increased faster and faster in the first lockdown period.

For the second lockdown, neither the best linear nor nonlinear models had a significant p-value for the risk perception term, which means the risk perception variable in the second lockdown was not crucial. The possible reason could be that the second lockdown took place between the 3-tier lockdown and Christmas holidays. There may have many uncertainties in estimating the risk perceptions and other psychological concerns. For example, some people might have avoided going outside because of the severe epidemic. However, other people might have insisted on going out to visit family members or shopping because of the release from restrictions expected for the holiday period.

As for the third lockdown period, the best linear and nonlinear models used the risk perception generated by Algorithm 3 EMZ_28. Even though the exponentially weighted moving z-score has a window size, it does not mean the method only takes N previous data points. However, it still considered all history data points, just decreasing weights. The good performance of EMZ_28 risk perception can be explained by local people judging the COVID-19 severity more realistically during the third lockdown than people were able to in the initial stage. That would be rational, as it had been a year at that time since the

first wave in the U.K., and people would have a few memories of the beginning stage and care more about the prevailing situation.

By comparing the results from research on the same topic, this study achieved similar conclusions in depicting the lockdown's exponentially diminishing impact on mobility. Joshi found that the drop rates of human mobility after lockdown implementation become slow as time went on [24]. Ganslmeier et al. [24] suggested that the non-compliance of lockdown results reflected by human mobility increased steadily during the first lockdown in the U.K. The current research incorporates the policy non-compliance and risk perceptions together, and the estimated coefficients through the Leeds case study agree with the observed phenomenon. In addition, the current research also discusses the changes in coefficients from the first lockdown to the third, representing the quantitative evolution of the population's risk perceptions and attitudes toward lockdowns.

## 4. Conclusions

The research explores the mobility changes caused by lockdown policy duration and dynamic risk perceptions. It is different from traditional survey-based research on risk perceptions. Our analysis provides three algorithms simulating the local public's daily risk perceptions toward COVID-19 using the official dashboard data. First, the time-varying z-score reflected the dynamic changes in risk perceptions as the epidemic developed in different stages. Moreover, the algorithm also considered the comprehensive information sources that might influence mobility, including local and national daily cases and deaths. Finally, the simulated results could capture the residents' risk perceptions toward the COVID-19 in the U.K., not simply proportional to the daily reported cases [21].

The case study was implemented in Leeds, which experienced three strict lockdown periods over a year. After tuning the hyper-parameters of different risk perception generation algorithms and comparing linear with nonlinear models, three selected nonlinear models respectively captured the mobility trends well in different lockdown periods. Since the model parameters are explicitly shown in the equation, the regression model could provide good interpretability. The social evolution phenomenon was exhibited by comparing the same parameter in different values of three lockdown periods. The model results reflect that the local population adapted to the lockdown lives in the COVID era. First, the strict lockdown policy displayed minor efficiency in restricting mobility as the number of lockdowns increased, which is consistent with global discoveries [4,5,39]. The last lockdown policy was less potent in restricting retail and recreation mobility than the first lockdown. In addition, the change in choice of dynamic risk perception algorithms could demonstrate that residents changed their attitudes toward the reported cases and deaths, indicating that residents tend to lose attention to all reported cases and focus on the recent epidemic developments. Taking Leeds as an example, during the most recent lockdown policy, the dynamic risk perceptions were estimated in an exponentially weighted way. Compared with the former model using all history information, the exponentially weighted approach is more rational, representing people focusing on their current lives rather than only being concerned about the epidemic. The regression coefficient of risk perceptions, which quantifies the negative influence on mobility, was lower by 40% in the third lockdown than in the first lockdown. The possible reason for this decline might be that promoting vaccination decreased the fatality rate, and people preferred to believe the pandemic was becoming harmless. The explanations of parameters are meaningful because they quantitatively present the existing phenomenon and could support deeper analysis. The policymakers could use the proposed approach to quickly evaluate the efficiency of lockdown policy when the next epidemic outbreaks. Additionally, the changes in risk perceptions and drop in mobility remind policymakers that lockdown may not be as effective over time, and that people can fall into lockdown fatigue. Governments could inform people with the latest knowledge on viruses and the pandemic, which would encourage the residents to have the right level of risk perceptions, and could trigger the population's self-protective behaviours and policy compliance when necessary. Furthermore, a resilient, healthy, inclusive, and safe

urban system should be considered in future urban planning and development [40]. The focus on jobs-housing balance, slow traffic, and improving accessibility to various services may all bring great benefits, not limited to pandemic control, to urban residents [41,42].

Local authorities and stakeholders could also put more investments into health and hygiene services and facilities in non-essential locations to cope with the epidemic. For example, retail and recreational places should have regular cleaning and ventilation. If necessary, in some policy contexts and countries, a vaccine passport policy [43] or proof of negative COVID-19 test could help prevent the virus spread in indoor activities [44].

However, this research still has certain limitations. First, the risk perception generation method assumes that people feel danger when cases or deaths increase, while the actual risk conceptions can be triggered by a comprehensive psychological process and personal social experience. Second, the model did not involve other COVID-19 dashboard data, such as the vaccination rate and the number of people in hospitals. Third, the vaccination project started with the third lockdown period; it may influence mobility and risk perceptions but was not considered in this study. Future work may focus on the statistical causal inference between each pair of factors and explore the mobility dynamics under the COVID-19 situation to investigate better public policy. To extend the model's feasibility, future research should aim to explore the methodology in other regions and discuss the risk perceptions and policy compliance in different cities and countries by comparing the coefficients and parameters in the regression. The GIS heatmap could be employed to illustrate the comparisons among regions in intuition and explore the spatiotemporal correlations between mobility and policy compliance.

**Author Contributions:** Conceptualisation, Sijin Wu and Susan Grant-Muller; methodology, Sijin; software, Sijin; validation, Sijin Wu and Susan Grant-Muller; formal analysis, Sijin; data curation, Sijin Wu; writing—original draft preparation, Sijin Wu; writing—review and editing, Sijin Wu, Susan Grant-Muller and Lili Yang; project administration, Susan Grant-Muller and Lili Yang; All authors have read and agreed to the published version of the manuscript.

**Funding:** This research is supported by the Shenzhen Science and Technology Special Project of the Epidemic (Grant No. JSGG20220301090202005).

**Institutional Review Board Statement:** Not applicable. The current research does not include the ethical concern and hence it does not require the ethical approval.

**Informed Consent Statement:** Not applicable. The current research utilized the available data from open-source, and the data does not include the individual information.

**Data Availability Statement:** The research data mentioned in this study is open-access, and here are links 1. Google LLC "*Google COVID-19 Community Mobility Reports*". https://www.google.com/covid19/mobility/ accessed on 30 October 2021. 2. https://coronavirus.data.gov.uk/ accessed on 10 March 2022.

**Acknowledgments:** I would like to express my deepest appreciation to Yuanxuan Yang, who provided generous and professional suggestions during the paper revision, making the manuscript targeted and fitting the journal's scope better.

**Conflicts of Interest:** The authors declare no conflict of interest.

## Appendix A.

*Appendix A.1. Feature Selection: Reducing Collinearity*

Data processing on variables alone is not enough to eliminate multicollinearity. Figure A1 shows that the policy time variable has high correlation coefficients with cumulative COVID-19 variables near to positive. This makes sense, because the cumulative features always increase over time.The solution is to eliminate the cumulative COVID-19 statistics and keep the variable 'policy time' only.

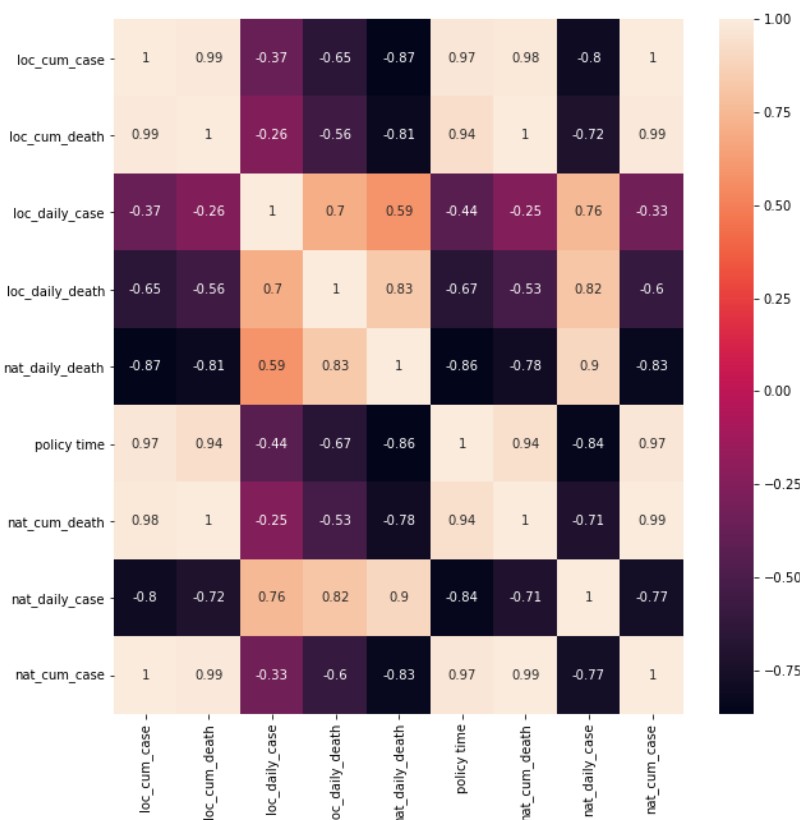

**Figure A1.** The correlation heatmap of independent variables.

*Appendix A.2. The Dynamic Risk Perceptions by Time-Varying z-Score and PCA*

The research has proposed three algorithms to simulate the risk perceptions, and each algorithm has several hyperparameters to choose from. This section lists the dynamic risk perception results of each algorithm.

There are two steps for generating the dynamic risk perception. The first step is using the proposed time-varying z-score algorithms for four sequences separately. For Algorithm 1, the z-score is calculated by using moving average and standard deviation. There are three alternatives for the window size: 7, 14 and 28, representing the length of public memory. As for Algorithm 2, there is no hyperparameter to tune, so only one result is generated. Similarly, Algorithm 3 also has a window size as the hyperparameter to tune even if it used all historical data. To fairly compare the effect of each algorithm, window size options are still 7, 14, and 28. Table A1 shows the names of these risk perceptions.

**Table A1.** The risk perceptions to examine in the case study (mz: moving z-score; emz: exponentially weighted moving z-score).

| Algorithm | Window Size | Risk Perceptions |
|---|---|---|
| | 7 | MZ_7 |
| Algorithm 1 | 14 | MZ_14 |
| | 28 | MZ_28 |
| Algorithm 2 | - | MZ_all |
| | 7 | EMZ_7 |
| Algorithm 3 | 14 | EMZ_14 |
| | 28 | EMZ_28 |

The visualisation results are divided into three lockdown periods to study and demonstrate in Figure A2. There are apparent gaps between risk curves generated by moving

z-score and exponentially weighted moving z-score. The EMZ curves are higher than the MZ ones no matter the window size at the middle of each lockdown period.

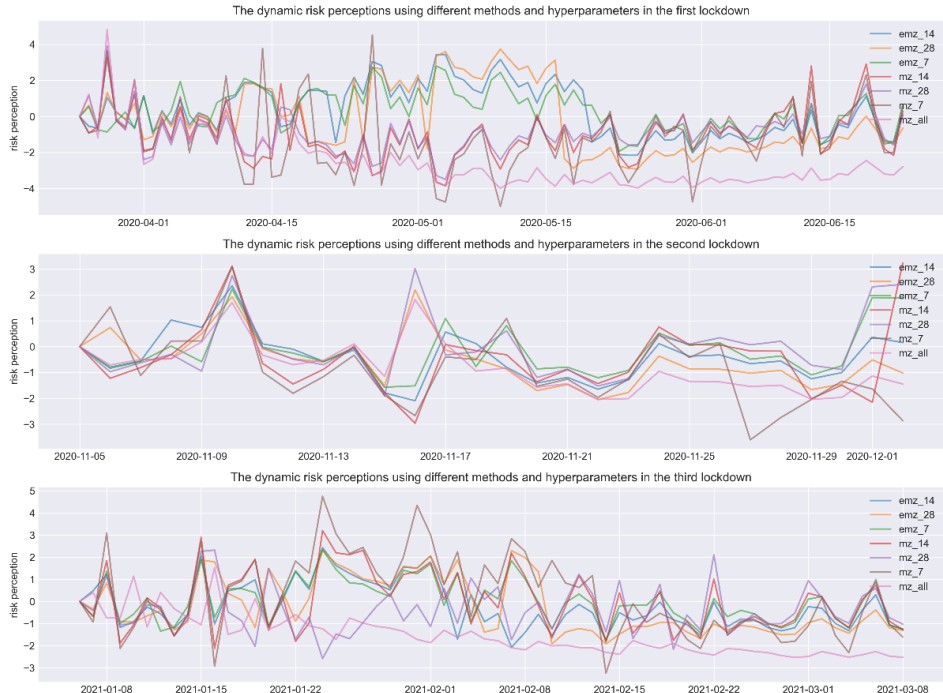

**Figure A2.** The dynamic risk perceptions generated by different methods in three lockdown periods.

The research tried to validate the simulated dynamic risk perceptions by comparing the general trends discovered in the U.K. [21]. In Figure A3, the left is from the Schneider et al. longitudinal survey results across the U.K. from March 2020 to January 2021. According to the mean value in each time point, the public's risk perception peaked in March 2020 and then decayed until September 2020, before increasing in January 2021, which was still lower than the initial level. The risk perception generated by the EMZ_28 method could capture the similar trend exactly shown on the right in Figure A3.

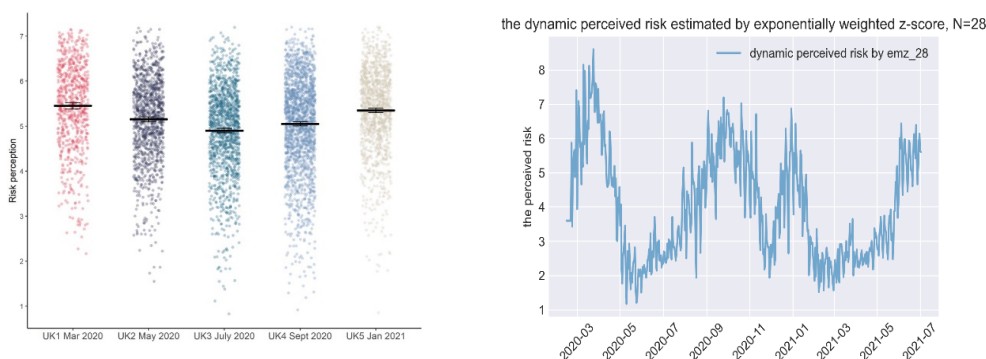

**Figure A3.** The risk perception comparison between survey results and emz_28.

*Appendix A.3. Experiment Results*

This subsection displays all experiment results mentioned in Section 3.3. Two tables record the estimated values of parameters and their goodness-of-fit for linear and nonlinear models, respectively. Figure A4 shows the experiment records of the nonlinear model for each lockdown period and the choice of algorithms in different hyper-parameters, where the highlight region is the best fitting choice for each lockdown period. Figure A5 is the summary of all experiments for the linear model. The complete excel files are saved as supporting materials.

**Figure A4 — First-lockdown (nonlinear model)**

| Model | N | Param | Params | T-test | P-value | Adj_R2 |
|---|---|---|---|---|---|---|
| exponentially weighted moving risk | 7 | c | -80.80 | -257.13 | 0.00 | Adj_R2:0.928804, MSE: 275.8 |
| | | β1 | 26.82 | 16.53 | 0.00 | |
| | | β2 | 0.04 | 98.74 | 0.00 | |
| | | α | 0.03 | 0.16 | 0.87 | |
| exponentially weighted moving risk | 14 | c | -80.81 | -254.62 | 0.00 | Adj_R2:0.9288283, MSE: 275.8 |
| | | β1 | 26.84 | 16.50 | 0.00 | |
| | | β2 | 0.04 | 95.71 | 0.00 | |
| | | α | 0.03 | 0.23 | 0.82 | |
| exponentially weighted moving risk | 28 | c | -80.74 | -284.16 | 0.00 | Adj_R2:0.9292953, MSE:273.9 |
| | | β1 | 26.71 | 16.74 | 0.00 | |
| | | β2 | 0.04 | 98.04 | 0.00 | |
| | | α | -0.09 | -0.79 | 0.43 | |
| moving risk | 7 | c | -81.06 | -249.99 | 0.00 | Adj_R2:0.9308883, MSE:267.8 |
| | | β1 | 27.31 | 16.96 | 0.00 | |
| | | β2 | 0.04 | 108.51 | 0.00 | |
| | | α | -0.16 | -1.63 | 0.11 | |
| moving risk | 14 | c | -81.14 | -230.50 | 0.00 | Adj_R2:0.930973, MSE:267.4 |
| | | β1 | 27.43 | 16.89 | 0.00 | |
| | | β2 | 0.04 | 106.25 | 0.00 | |
| | | α | -0.24 | -1.66 | 0.10 | |
| moving risk | 28 | c | -80.81 | -209.02 | 0.00 | Adj_R2:0.9287917, MSE:275.9 |
| | | β1 | 26.83 | 15.74 | 0.00 | |
| | | β2 | 0.04 | 101.17 | 0.00 | |
| | | α | -0.02 | -0.10 | 0.92 | |
| moving risk | All | c | -82.00 | -188.55 | 0.00 | Adj_R2:0.9376691, MSE:241.5015 |
| | | β1 | 29.56 | 17.46 | 0.00 | |
| | | β2 | 0.03 | 103.51 | 0.00 | |
| | | α | -0.51 | -3.52 | 0.00 | |

**Figure A4 — Second-Lockdown (nonlinear model)**

| Model | N | Param | Params | T-test | P-value | Adj_R2 |
|---|---|---|---|---|---|---|
| exponentially weighted moving risk | 7 | c | -62.22 | -44.76 | 0.00 | Adj_R2:0.7969454, MSE:279.4 |
| | | β1 | 35.67 | 9.35 | 0.00 | |
| | | β2 | 0.06 | 3.97 | 0.00 | |
| | | α | -0.02 | -0.02 | 0.98 | |
| exponentially weighted moving risk | 14 | c | -62.06 | -46.08 | 0.00 | Adj_R2:0.7997176, MSE:275.6 |
| | | β1 | 35.42 | 9.35 | 0.00 | |
| | | β2 | 0.06 | 4.26 | 0.00 | |
| | | α | 0.40 | 0.58 | 0.57 | |
| exponentially weighted moving risk | 28 | c | -62.17 | -46.92 | 0.00 | Adj_R2:0.7980499, MSE:277.9 |
| | | β1 | 35.52 | 9.34 | 0.00 | |
| | | β2 | 0.07 | 4.25 | 0.00 | |
| | | α | 0.27 | 0.37 | 0.72 | |
| moving risk | 7 | c | -62.06 | -48.24 | 0.00 | Adj_R2:0.8050882, MSE:268.2 |
| | | β1 | 35.20 | 9.44 | 0.00 | |
| | | β2 | 0.07 | 5.12 | 0.00 | |
| | | α | 0.52 | 1.01 | 0.32 | |
| moving risk | 14 | c | -62.02 | -45.08 | 0.00 | Adj_R2:0.7992887, MSE:276.2 |
| | | β1 | 35.47 | 9.35 | 0.00 | |
| | | β2 | 0.06 | 3.96 | 0.00 | |
| | | α | 0.26 | 0.53 | 0.60 | |
| moving risk | 28 | c | -62.30 | -46.22 | 0.00 | Adj_R2:0.7976699, MSE:278.4 |
| | | β1 | 35.74 | 9.44 | 0.00 | |
| | | β2 | 0.06 | 4.21 | 0.00 | |
| | | α | -0.16 | -0.30 | 0.77 | |
| moving risk | All | c | -62.23 | -45.72 | 0.00 | Adj_R2:0.7978839, MSE:278.1 |
| | | β1 | 35.64 | 9.37 | 0.00 | |
| | | β2 | 0.06 | 2.85 | 0.01 | |
| | | α | -0.29 | -0.34 | 0.73 | |

**Figure A4 — Third-Lockdown (nonlinear model)**

| Model | N | Param | Params | T-test | P-value | Adj_R2 |
|---|---|---|---|---|---|---|
| exponentially weighted moving risk | 7 | c | -68.81 | -205.31 | 0.00 | Adj_R2:0.8426589, MSE:137 |
| | | β1 | 9.81 | 6.68 | 0.00 | |
| | | β2 | 0.04 | 45.33 | 0.00 | |
| | | α | -0.02 | -0.08 | 0.94 | |
| exponentially weighted moving risk | 14 | c | -68.84 | -216.08 | 0.00 | Adj_R2:0.8442878, MSE:135.6 |
| | | β1 | 9.84 | 6.81 | 0.00 | |
| | | β2 | 0.04 | 47.02 | 0.00 | |
| | | α | 0.16 | 0.78 | 0.44 | |
| exponentially weighted moving risk | 28 | c | -68.69 | -213.16 | 0.00 | Adj_R2:0.8502894, MSE:130.4 |
| | | β1 | 9.55 | 6.69 | 0.00 | |
| | | β2 | 0.04 | 41.17 | 0.00 | |
| | | α | -0.30 | -1.72 | 0.09 | |
| moving risk | 7 | c | -68.75 | -194.97 | 0.00 | Adj_R2:0.8431947, MSE:136.6 |
| | | β1 | 9.70 | 6.56 | 0.00 | |
| | | β2 | 0.04 | 43.10 | 0.00 | |
| | | α | -0.06 | -0.45 | 0.65 | |
| moving risk | 14 | c | -68.77 | -210.94 | 0.00 | Adj_R2:0.8438013, MSE:136 |
| | | β1 | 9.72 | 6.68 | 0.00 | |
| | | β2 | 0.04 | 46.66 | 0.00 | |
| | | α | -0.10 | -0.66 | 0.51 | |
| moving risk | 28 | c | -68.93 | -211.99 | 0.00 | Adj_R2:0.8470236, MSE:133.2 |
| | | β1 | 9.96 | 6.94 | 0.00 | |
| | | β2 | 0.04 | 50.23 | 0.00 | |
| | | α | -0.23 | -1.29 | 0.20 | |
| moving risk | All | c | -69.40 | -140.70 | 0.00 | Adj_R2:0.8486247, MSE:131.8 |
| | | β1 | 10.84 | 6.89 | 0.00 | |
| | | β2 | 0.04 | 35.37 | 0.00 | |
| | | α | -0.48 | -1.50 | 0.14 | |

**Figure A4.** The nonlinear model results of each generated risk perceptions in three lockdown periods.

**Figure A5 — First-lockdown (linear model)**

| Model | N | Param | Params | T-test | P-value | Adj_r2 |
|---|---|---|---|---|---|---|
| exponentially weighted moving risk | 7 | c | -85.99 | -146.08 | 0.00 | 0.91 |
| | | β1 | 37.48 | 18.50 | 0.00 | |
| | | β2 | 0.26 | 26.78 | 0.00 | |
| | | α | -0.15 | -0.77 | 0.44 | |
| exponentially weighted moving risk | 14 | c | -85.86 | -144.46 | 0.00 | 0.91 |
| | | β1 | 37.22 | 18.35 | 0.00 | |
| | | β2 | 0.26 | 26.06 | 0.00 | |
| | | α | -0.19 | -1.21 | 0.23 | |
| exponentially weighted moving risk | 28 | c | -85.99 | -156.39 | 0.00 | 0.91 |
| | | β1 | 37.51 | 18.99 | 0.00 | |
| | | β2 | 0.26 | 26.68 | 0.00 | |
| | | α | -0.15 | -1.24 | 0.22 | |
| moving risk | 7 | c | -85.86 | -157.41 | 0.00 | 0.91 |
| | | β1 | 37.25 | 19.01 | 0.00 | |
| | | β2 | 0.27 | 29.72 | 0.00 | |
| | | α | 0.21 | 1.92 | 0.06 | |
| moving risk | 14 | c | -85.75 | -147.62 | 0.00 | 0.91 |
| | | β1 | 37.08 | 18.60 | 0.00 | |
| | | β2 | 0.26 | 28.94 | 0.00 | |
| | | α | 0.27 | 1.71 | 0.09 | |
| moving risk | 28 | c | -85.36 | -141.47 | 0.00 | 0.91 |
| | | β1 | 36.09 | 17.86 | 0.00 | |
| | | β2 | 0.26 | 28.75 | 0.00 | |
| | | α | 0.52 | 2.59 | 0.01 | |
| moving risk | All | c | -85.24 | -168.53 | 0.00 | 0.93 |
| | | β1 | 35.27 | 19.34 | 0.00 | |
| | | β2 | 0.30 | 29.26 | 0.00 | |
| | | α | 0.89 | 4.92 | 0.00 | |

**Figure A5 — Second-lockdown (linear model)**

| Model | N | Param | Params | T-test | P-value | Adj_r2 |
|---|---|---|---|---|---|---|
| exponentially weighted moving risk | 7 | c | -60.85 | -27.58 | 0.00 | 0.76 |
| | | β1 | 34.81 | 7.65 | 0.00 | |
| | | β2 | 0.11 | 1.02 | 0.32 | |
| | | α | 0.16 | 0.23 | 0.82 | |
| exponentially weighted moving risk | 14 | c | -60.76 | -27.99 | 0.00 | 0.76 |
| | | β1 | 34.74 | 7.70 | 0.00 | |
| | | β2 | 0.12 | 1.06 | 0.30 | |
| | | α | 0.31 | 0.62 | 0.54 | |
| exponentially weighted moving risk | 28 | c | -60.96 | -27.84 | 0.00 | 0.76 |
| | | β1 | 34.91 | 7.68 | 0.00 | |
| | | β2 | 0.12 | 1.06 | 0.30 | |
| | | α | -0.02 | -0.04 | 0.97 | |
| moving risk | 7 | c | -61.09 | -28.31 | 0.00 | 0.76 |
| | | β1 | 34.87 | 7.74 | 0.00 | |
| | | β2 | 0.15 | 1.24 | 0.23 | |
| | | α | 0.34 | 0.63 | 0.53 | |
| moving risk | 14 | c | -60.76 | -27.99 | 0.00 | 0.76 |
| | | β1 | 34.74 | 7.70 | 0.00 | |
| | | β2 | 0.12 | 1.06 | 0.30 | |
| | | α | 0.31 | 0.62 | 0.54 | |
| moving risk | 28 | c | -60.96 | -27.84 | 0.00 | 0.76 |
| | | β1 | 34.91 | 7.68 | 0.00 | |
| | | β2 | 0.12 | 1.06 | 0.30 | |
| | | α | -0.02 | -0.04 | 0.97 | |
| moving risk | All | c | -60.48 | -27.34 | 0.00 | 0.76 |
| | | β1 | 34.21 | 7.51 | 0.00 | |
| | | β2 | 0.05 | 0.37 | 0.72 | |
| | | α | -0.74 | -0.82 | 0.42 | |

**Figure A5 — Third-lockdown (linear model)**

| Model | N | Param | Params | T-test | P-value | Adj_r2 |
|---|---|---|---|---|---|---|
| exponentially weighted moving risk | 7 | c | -71.17 | -122.23 | 0.00 | 0.82 |
| | | β1 | 14.92 | 8.61 | 0.00 | |
| | | β2 | 0.22 | 15.36 | 0.00 | |
| | | α | -0.16 | -0.70 | 0.49 | |
| exponentially weighted moving risk | 14 | c | -71.51 | -125.77 | 0.00 | 0.82 |
| | | β1 | 15.47 | 9.11 | 0.00 | |
| | | β2 | 0.23 | 15.80 | 0.00 | |
| | | α | 0.26 | 1.22 | 0.23 | |
| exponentially weighted moving risk | 28 | c | -70.90 | -122.00 | 0.00 | 0.83 |
| | | β1 | 14.47 | 8.49 | 0.00 | |
| | | β2 | 0.21 | 13.98 | 0.00 | |
| | | α | -0.33 | -1.80 | 0.08 | |
| moving risk | 7 | c | -70.95 | -118.81 | 0.00 | 0.82 |
| | | β1 | 14.57 | 8.43 | 0.00 | |
| | | β2 | 0.22 | 14.86 | 0.00 | |
| | | α | -0.17 | -1.40 | 0.17 | |
| moving risk | 14 | c | -71.16 | -126.03 | 0.00 | 0.82 |
| | | β1 | 14.91 | 8.72 | 0.00 | |
| | | β2 | 0.22 | 15.85 | 0.00 | |
| | | α | -0.16 | -1.00 | 0.32 | |
| moving risk | 28 | c | -71.41 | -129.48 | 0.00 | 0.82 |
| | | β1 | 15.31 | 9.07 | 0.00 | |
| | | β2 | 0.23 | 16.87 | 0.00 | |
| | | α | -0.22 | -1.14 | 0.26 | |
| moving risk | All | c | -71.18 | -127.29 | 0.00 | 0.82 |
| | | β1 | 14.91 | 8.74 | 0.00 | |
| | | β2 | 0.24 | 11.32 | 0.00 | |
| | | α | 0.43 | 1.04 | 0.30 | |

**Figure A5.** The linear model results of each generated risk perceptions in three lockdown periods.

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
