# Peer review of "Modelling the Mobility Changes Caused by Perceived Risk and Policy Efficiency"

_ijgi, doi:10.3390/ijgi11080453_

Round 1

Reviewer 1 Report

I accept the majority your remarks and corrections, your manuscript has improved significantly. But the resolution of most of your figures is still low, and the background of Figure 2. (p. 8.) is very dark, so I assume that it is not easy to interpret this map visually. 

Author Response

Comment #. "I accept the majority your remarks and corrections, your manuscript has improved significantly. But the resolution of most of your figures is still low, and the background of Figure 2. (p. 8.) is very dark, so I assume that it is not easy to interpret this map visually. "

Reply: Thanks for your constructive advice again. We have tried several solutions to figure out to display the images in the manuscripts better.

First, we re-plot images and save them, choosing dpi=1000.

Second, we found the issue of low resolution may come from an option in Word. So, we have closed the auto-compress of images in Word and reloaded the images in the manuscripts.

Finally, if the above steps still do not produce a satisfactory resolution, we will upload the supporting files containing all original pictures through the submission system, and the editor may have better solutions to display them.

Particularly, figure 2 is to display the distribution of retail and recreational Points of Interest in Leeds. The base map is chosen as black to highlight the points. The remade map in the revised manuscript should now be clearer.

Reviewer 2 Report

The revised version of the manuscript has fixed some issues while also introducing new ones. For instance, the abbreviations have been fixed throughout the text, and some information from the appendices has been integrated into the main text, improving its structure and readability. At the same time, however, the appendices still contain information relevant to the presented research, to the point that parts of them (in particular, figures 11 and 12 from appendix A3) are now directly referenced in the main text. Again, appendices should be used for non-essential and supplementary information, and all information necessary for analysis of the presented research should be part of the main text. In this context, please consider either removing the references to figures 11 and 12, or integrating these figures into the main text. In addition, I believe that the appendices should be numbered in the same order in which they are referenced in the text. Currently, the appendix is divided into three parts (A1, A2 and A3), where the second part is referenced first (in line 202), before the first part (A1 is referenced in line 329), and only the third part is referenced in the correct order (in line 387).

Finally, there still remains a number of broken figure and literature references, see eg.:
- line 393
- line 401
- line 410

As well as some minor text issues:
Lines 462-463: "because of" is doubled
Line 509: "mobilityremind"

On the whole, while the text has been noticeably improved, I believe that the remaining issues should be fixed before it can be considered for publication.

Author Response

Comment #1: "The revised version of the manuscript has fixed some issues while also introducing new ones. For instance, the abbreviations have been fixed throughout the text, and some information from the appendices has been integrated into the main text, improving its structure and readability. At the same time, however, the appendices still contain information relevant to the presented research, to the point that parts of them (in particular, figures 11 and 12 from appendix A3) are now directly referenced in the main text. Again, appendices should be used for non-essential and supplementary information, and all information necessary for analysis of the presented research should be part of the main text. In this context, please consider either removing the references to figures 11 and 12, or integrating these figures into the main text."

Reply: Thanks for your careful review of the manuscript. Figure 11 and figure 12 are used to display all experiment results, which are stored in the csv files and will be uploaded through the submission system. So we authors think it is better to remove their references in the main text and replace them with "see Appendix" on lines 402 and 413.

Comment #2. "In addition, I believe that the appendices should be numbered in the same order in which they are referenced in the text. Currently, the appendix is divided into three parts (A1, A2 and A3), where the second part is referenced first (in line 202), before the first part (A1 is referenced in line 329), and only the third part is referenced in the correct order (in line 387)."

Reply: We are sorry that the order is misleading due to carelessness in original line 202. It should be A1 rather than A2. And the logical order of the Appendix sections is now consistent with the main text.

Finally, there still remains a number of broken figure and literature references, see eg.:
- line 393
- line 401
- line 410

As well as some minor text issues:
Lines 462-463: "because of" is doubled
Line 509: "mobilityremind"
Reply: Thanks for pointing these out,  the above issues are fixed in the revised manuscript.

Reviewer 3 Report

This paper explores the mobility changes caused by three lockdown policies and dynamic risk perceptions in Leeds using Google’s mobility index data and official daily case reports. They used time-varying z-scores and Principal Component Analysis (PCA) to estimate the dynamic daily risk receptions of local people based on multi-dimensional daily COVID-19 reports. Then, they adopted nonlinear regression model to fit mobility changes. The topic of this work interesting, but I do not think that the authors have addressed their research questions well. I have to point out some problems of this paper before I can recommend it to be published.

1.        The introduction section needs to be reorganized. At this present version, it is not clear for readers to follow the author’s motivations. For example, in the first paragraph, the author puts forward the purpose of this article only according to some social backgrounds. The highlighted text (line 60-67) in the second paragraph is incoherent.

2.       The literature about the relationship between mobility and risk perception that are repeated in the second and third paragraphs. This part also needs to be modified.

3.       The two research gaps are not clear and need to be reorganized.

4.       The authors stated that they try to answer the causality between mobility and risk perceptions in the introduction, but I don’t find relevant  descriptions in the part of model results and discussion.

5.       Although the dynamic model results fit well, how does the nonlinear regression models answer the causal relationship between mobility changes and risk perception when they have assumed the relationship of lockdown policy and risk perceptions and mobility changes? I don't find any counterfactual statements in the part of research design.

6.       The authors should perhaps compare with similar research in the discussion of results.

Author Response

Thanks for your specific reviews and advice. We have made the point-by-point response. 

Please see the attached response letter and revised manuscript.

Round 2

Reviewer 3 Report

The authors revised the paper based on my comments. This version is a significant improvement over the first submission. Thus, I appreciate their attention to most of my suggestions and consider the paper to be ready for pubilication. Of course, they need to add some literature in the conclusion to response their findings and policy recommendations.

Author Response

Thanks for your recognition of our last revision. And yes, we should add the literature in the conclusion section supporting the findings and policy recommendations.

So in line 515, we reviewed that our risk perception algorithms could reach similar results with longitudinal surveys.

"Finally, the simulated results could capture the residents' risk perceptions toward the COVID-19 in the U.K., not simply proportional to the daily reported cases[21]."

Moreover, in lines 523-525, the regression results' interpretations align with other related research conclusions.

"First, the strict lockdown policy displayed minor efficiency in restricting mobility as the number of lockdowns increased, which is consistent with global discoveries[4], [5], [39]."

Finally, we added the literature references in lines 550 and 555, stating that our suggested policies are considered and discussed in related research.

" Furthermore, a resilient, healthy, inclusive and safe urban system should be considered in future urban planning and development[40]. The focus on jobs-housing balance, slow traffic, and improving accessibility to various services may all bring great benefits, not limiting to pandemic control, to urban residents[41], [42]."

"If necessary, in some policy contexts and countries, a vaccine passport policy[43] or proof of negative COVID test could help prevent the virus spread in indoor activities[44]"

In the revised manuscript, we have used green to highlight the above adjustments. The newly updated references are listed below.

[40]         R. L. Abduljabbar, S. Liyanage, and H. Dia, ‘The role of micro-mobility in shaping sustainable cities: A systematic literature review’, Transportation Research Part D: Transport and Environment, vol. 92, p. 102734, Mar. 2021, doi: 10.1016/j.trd.2021.102734.

[41]         D. M. Harrington and M. Hadjiconstantinou, ‘Changes in commuting behaviours in response to the COVID-19 pandemic in the UK’, Journal of Transport & Health, vol. 24, p. 101313, Mar. 2022, doi: 10.1016/j.jth.2021.101313.

[42]         Y. Liu, L. C. Tong, X. Zhu, and W. Du, ‘Dynamic activity chain pattern estimation under mobility demand changes during COVID-19’, Transportation Research Part C: Emerging Technologies, vol. 131, p. 103361, Oct. 2021, doi: 10.1016/j.trc.2021.103361.

[43]         R. K. Goel and J. R. Jones, ‘Managing the risk of COVID‐19 via vaccine passports: Modeling economic and policy implications’, MDE Manage Decis Econ, p. 10.1002/mde.3546, Jan. 2022, doi: 10.1002/mde.3546.

[44]         D. Michaels, E. J. Emanuel, and R. A. Bright, ‘A National Strategy for COVID-19: Testing, Surveillance, and Mitigation Strategies’, JAMA, vol. 327, no. 3, pp. 213–214, Jan. 2022, doi: 10.1001/jama.2021.24168.

This manuscript is a resubmission of an earlier submission. The following is a list of the peer review reports and author responses from that submission.

Round 1

Reviewer 1 Report

I like the idea of the manuscript, your research is very current and relevant. However, there are some shortcomings in the manuscript that need to be corrected.

I think the research is too much focused on Leeds, even though the IJGI’s readership is global. How much and how do you think your Leeds research can be generalised to other cities in England, for example?

Your research often uses the COVID-19 Mobility Report as a mobility benchmark, but, as I suppose, it makes less mention that the mobility value measured by the Report may change not only because of the consequences of COVID-19 but also, in my opinion, partly because of other factors, as well. According to the authors, can the effects of these other factors be separated from the effects of COVID-19 factors in your analysis, and if so, how?

I think that the authors should better analyse the existing literature on the GIS-based use of the COVID-19 Mobility Report.

Despite the wide variety of quantitative analysis results, the summary section seems to be short, I recommend expanding it extensively. And a more structured summary of the main findings of the study should be summarised here, as well. If possible, there could be also 1-2 practical suggestions (e. g. in urban development or health policy) on your quantitative findings here.

In terms of formal points, many illustrations (e. g. charts) are very small in size, making them difficult to read. Therefore, I think it is necessary to increase their size (even by placing them in the Annex by the authors).  The text contains edit errors in some places (e. g. "Error! Reference source not found", p. 8.), and they need to be fixed. In the Data Availability Statement (p. 12), it is recommended to indicate download dates.

Overall, the study has a relevant and novel scientific approach, but it contains errors in content and form of varying severity that need to be corrected.

Reviewer 2 Report

The paper attempts to model the global and local changes in human mobility in relation to the enforcement of lockdown policies as well as the evolving perception of COVID-19 risk. Citizen mobility has been obtained from Google Community Mobility Report for retail and recreational facilities for the city of Leeds, while the daily numbers of COVID-19 cases and deaths in Leeds as well as in the whole of UK have been used as input for simulating the change in risk perception. The mobility report data contains information about non-essential retail and recreational facilities (which were closed during lockdown), however it lacks information on supermarkets (which were open). Thus, while it may be used to gauge the effectiveness of lockdown policies (eg. on recreational facilities), it cannot be used to assess the change in everyday shopping behaviour.

The authors chose to use PCA and a time-varying z-score to simulate the perception of pandemic severity through multi-dimensional daily reports and history information. The time-varying z-score is computed using three different methods: the moving average with specific window size (abbreviated "mz_window_size"), the moving average with all history information (abbreviated "mz_all") and the exponentially weighted moving average (abbreviated "emz_window_size"). The chosen window sizes represent citizen risk memory in time periods of 7, 14 and 28 days. The z-score is generated on the basis of daily local and national COVID-19 cases and deaths, and PCA is used to reduce the four time series into one comprehensively perceived risk. The association between mobility changes and independent variables in the form of policy duration and dynamic risk perception is then tested with the use of linear and non-linear regression models, with the non-linear models achieving higher fitting scores. The results indicate that while the population mobility was affected by long-term history of COVID-19 cases during the first two lockdowns, short-term data was primarily used in consideration of mobility during the third lockdown. The results have been validated by comparison to general trends of risk perception in the UK established through citizen surveys by Schneider et al (2021).

My main issue with the paper is its structure, as a lot of important information (such as data preprocessing, feature selection and result validation) has been moved to the appendix. Appendices are generally used for non-essential reading and supplementary information. Currently it is not possible to appreciate the presented research without reading the appendices, which is why I believe that their content should be merged with the main text. Moreover, all abbreviations should be properly explained before they are used. Currently, the text defines acronyms such as MZ, EMZ and EWM, however the provided images and formulas contain abbreviations such as EMV, EMA and EWMV, which have not been described anywhere. In addition, the paper contains a number of broken figure and literature references as well as minor grammatical errors (see detailed recommendations below). Finally, I also have an issue with the paper's opening paragraph, which contains general statements regarding COVID-19 transmission rates and prevention policies. While this information could be considered common knowledge, they should be backed up by appropriate references (eg. WHO reports).

Detailed recommendations:

Lines 35-51: Although these statements could be considered common knowledge, scientific accuracy requires them to be backed up by appropriate references, which may be obtained eg. from WHO reports.

Line 65: "areas (...) ARE positively associated"
Line 75: I believe "changes in risk perception" sounds less confusing.
Line 76: do you mean "beginning at the start of the pandemic"?
Line 123: "quantitatively"
Line 132: "perceived risk" or "risk perception"
Line 155: again, "perceptions perceived" is not a good expression
Line 290: "Error! Reference source not found."
Line 342: "Error! Reference source not found."
Line 350: "Error! Reference source not found."
Line 360: "Error! Reference source not found."
Line 543: "validate"
Line 548: "emv_28" has not been properly defined in the text
Figure A9: the image to the right references an undefined method "ewmv_28"

Reviewer 3 Report

Review of Manuscript IJGI 1655001
Title: Modelling the mobility changes caused by perceived risk and policy efficiency: a case study in Leeds

Review Text:
The paper describes an approach to understand the lockdown regimes and their effect on human mobility in Leeds. The authors try to model the local restrictions and risk perceptions and their mobility behavior - from March 2020 to March 2021. Basically, the paper follows an interesting approach by a local time-varying z-score, which is adopted from economic research.  
The introduction and the relevant literature are well written and cover the current relevant scientific publications in that particular field. Also, the research questions and objectives are clearly documented for the reader.
As stated before, the paper and the approach would be generally interesting, but the execution and the methodological set-up is rather poor. As the methodology does not contain any geographic/spatial component (other than being applied to Leeds), I wonder why the authors would like to publish in the the IJ of Geoinformation. Especially, the algorithmic part is truly a-spatial and could be published in any other journal. If the authors would have reported on a more spatial methodology and/or worked with any kind of spatial definiton of "local" perceived risk, I would be happy. But this is not the case - which leaves me with the impression that the paper does not fit the objectives of the journal at all. 

some minor issues: 
- several spaces are missing (especially before/after paranthesis)
- several references are not found and thus not correctly cited.